# Dose Effect of Milk Thistle (*Silybum marianum*) Seed Cakes on the Digestibility of Nutrients, Flavonolignans and the Individual Components of the Silymarin Complex in Horses

**DOI:** 10.3390/ani11061687

**Published:** 2021-06-05

**Authors:** Hana Dockalova, Ladislav Zeman, Daria Baholet, Andrej Batik, Sylvie Skalickova, Pavel Horky

**Affiliations:** Department of Animal Nutrition and Forage Production, Mendel University in Brno, Zemědělská 1, 61300 Brno, Czech Republic; zeman@mendelu.cz (L.Z.); daria.baholet@mendelu.cz (D.B.); andrej.batik@mendelu.cz (A.B.); sylvie.skalickova@gmail.com (S.S.); pavel.horky@mendelu.cz (P.H.)

**Keywords:** silymarin, horse nutrition, blood, plasma, silybin, silydianin, silychristin

## Abstract

**Simple Summary:**

*Silybum marianum* is a well-known herb in terms of its pharmacological activities, and it is used as both a medicament and a dietary supplement (phytobiotics). Milk thistle seeds contain a mixture of flavonoids known as silymarin, which consists of silybin, isosilybin, silychristine, and silydianin. Until now, there has been no evidence of monitoring the digestibility of silymarin complex in horses. The aim of the research was to evaluate digestibility of silymarin complex and the effect of nutrient digestibility in horses. Different daily feed doses of milk thistle expeller (0 g, 100 g, 200 g, 400 g, 700 g) were administered to five mares kept under the same conditions and at the same feed rations. We monitored the digestibility of silymarin, digestible energy, crude protein, crude fat, crude fiber, nitrogen-free extract, crude ash, calcium, phosphorus, and plasma profile. Statistically significant differences (*p* ≤ 0.05) were found between daily doses in digestibilities of flavonolignans and nutrients. Our findings showed the digestibility of flavonolignans increased with the daily dose and then stagnated with the dose of milk thistle seed cakes at 700 g/day.

**Abstract:**

Milk thistle seeds contain a mixture of flavonoids known as silymarin, which consists of silybin, isosilybin, silychristine, and silydianin. Until now, there has been no evidence of monitoring the digestibility of silymarin complex in horses. The aim of the research was to evaluate the digestibility of silymarin complex and the effect of nutrient digestibility in horses. Different daily feed doses (FD) of milk thistle expeller (0 g, 100 g, 200 g, 400 g, 700 g) were administered to five mares kept under the same conditions and at the same feed rations. Digestibility of silymarin complex was monitored by HPLC-UV. Digestible energy (DE), crude protein, crude fat, crude fiber, nitrogen-free extract (NFE), crude ash, calcium (Ca), and phosphorus (P) were determined according ISO/IEC 17025:2017. The biochemical profile of blood plasma (total protein, albumin, alanine transaminase (ALT), aspartate transaminase (AST), alkaline phosphatase (ALP), gamma-glutamyl transferase (GGT), bilirubin, total cholesterol, HDL cholesterol, LDL cholesterol, triacyl glyceride (TAG), non-esterified fatty acid (NEFA), creatine kinase (CK), creatinine, urea, glutathione peroxidase (GSH-Px), total antioxidant status (TAS), glucose, calcium, and inorganic phosphate) was investigated. Moreover, the flavonolignans of the silymarin complex in plasma were detected. Statistically significant differences (*p* ≤ 0.05) were found between daily doses of milk thistle expellers in digestibilities. Our findings showed the digestibility of flavonolignans increased with the daily dose and then stagnated with the dose of milk thistle seed cakes at 700 g/day.

## 1. Introduction

The feeding of milk thistle (*Silybum marianum*) to horses has recently become a popular phenomenon because it is a well-known herb for its pharmacological activities and is used as both a medicament and a dietary supplement (phytobiotics). Milk thistle evinces antioxidative, antimicrobial, anticancer, neuroprotective, anti-diabetic, cardiovascular-protective, and other effects [1]. Its protective effect is connected to different mechanisms, such as repressing toxin penetration into hepatic cells, increasing antioxidant activity, decreasing lipid peroxidation, and enhancing hepatocyte protein synthesis. The estimated positive effects can be clarified on the basis of its antioxidant properties due to the phenolic nature of its flavonolignans [2]. On the contrary, the weakness of some phenolic compounds is thus far insufficient in terms of information about their metabolism in the organism [3]. 

The mixture of flavonolignans known as silymarin complex represents about 1.5–3% of the drug dry matter and consists of silybin or silibinin (50–60%), isosilybin (≈15%), silychristin (≈20%), and silydianin (≈10%) and other compounds (silimonin, isosilychristin, isosilybin) [4]. Up to 23 variants of flavonolignans were isolated in *S. marianum* [5]. The most attention has been given to silybin; however, other components are often ignored. This effect results in reproducibility as well as the exact composition of silymarin complex, which is often unknown and may vary depending on plant growth conditions, processing, etc. A comparison of detailed flavonolignan profiles has also not been available in different products thus far. The oil fraction, containing linoleic, oleic, palmitic, sterols, tocopherol, and phospholipids, is neglected components of *S. marianum* due to the focus on flavonolignans. Together with other substances such as betaine, apigenin, silybonol, and specific proteins, it can also contribute to the positive effects in addition to silymarin [6]. 

Silymarin has been considered as a dietary supplement recommended for dogs, cats, and horses with liver disease. However, the studies in these animal species are limited [7]. According to recent research results, feeding of milk thistle seed cakes can have a positive effect on the health of the sport horses [8]. The main drawback of milk thistle as feed additive is high variability due to the variable content of effective compounds, which is dependent on growth conditions [9]. 

In horses, silymarin complex is recommended for treatment of laminitis. The proposed mechanism of action is neutralization of endotoxins and reduction of lipopolysaccharides inducing lamellar separation due to the antioxidant and anti-inflammatory effects in an in vitro laminitis model [10]. Silymarin is also reported to increase glutathione levels in the blood of horses [11]. 

After oral administration, silybin undergoes extensive entero-hepatic circulation. The elimination half-life of silybin by humans is approximately 6 h. It has been established that 3–8% of orally administrated silybin is excreted in an unchanged form in the urine. About 80% of silybin is excreted as glucuronide and sulfate conjugates with bile. It is assumed that 20–40% of bile silybin is recovered, whereas the remaining part is excreted via feces. Silybin undergoes both phase I and phase II of biotransformation in liver cells [12]. Silybin digestibility in the gastrointestinal tract and bioavailability in the systemic circulation are dependent on various factors, such as the concentration of the preparation and the presence of additional substances with a solubilizing character (e.g., fat, proteins, amino acids, cholesterol, or other flavonoids) [12]. 

The aim of this study was to monitor the silymarin digestibility and its main subcomponents in various daily doses of milk thistle seed cakes in the form of a granular mixture with barley. The health status was also monitored by plasma biochemical parameters. The efficiency of silymarin digestibility has not been described in horse nutrition according to other authors thus far.

## 2. Materials and Methods 

### 2.1. Feed Processing and Diet Formulation

Different daily doses of milk thistle seed cakes (up to 0 g/day, 100 g/day, 200 g /day, 400 g/day, 700 g/day) were fed in the form of a granulated mixture with extruded barely in the experiment. The scheme of feed doses for horses is shown in Table 1. Nutrient composition of feeds forming the basic feed dose of horses is shown in Table 2, and nutrient composition of feed doses with different proportions of milk thistle seed cakes is shown in Table 3.

The daily intake of crude protein, crude fat, crude fiber, crude ash, Ca, and P was increased with a gradual increase in the feed of milk thistle seed cakes. NFE content of dry matter was detected in indirect proportion to the dose of milk thistle seed cakes.

### 2.2. Experimantal Animals, Housing, and Feeding Regimen

Five horses of the Czech Warmblood breed were included in the experiment for a monitored period of 15 weeks. The horses were dewormed 30 days before the beginning of the experiment. The mares were of various ages (8.1 ± 6.6 years) and weights (480 ± 58 kg); they had the optimal moderate body condition score (BCS) of 5 [13] and were not pregnant. The mares were clinically healthy (according to plasma profile, among others). The mares came from the same family. The housing was in the form of individual boxes. The mares went to the paddock every day for 8 h. The experiment took place in the period from October to mid-January, i.e., in the period with no available grazing in the paddock. All the mares received the same feed dose shown in Table 1, and the daily feeding dose was divided to two doses (morning and evening). The daily nutrient requirements were meted NRC [14]. Water was available ad libitum from automatic drinkers. The Committee of Experts for Ensuring the Welfare of Animals of the Mendel University in Brno approved the experiment protocol.

### 2.3. Analysis of Feed and Feces

Feed and feces samples were taken on days 20, 41, 62, 83, and 104 and were dried to 59 °C for 24 h before analyses. The dry matter was determined according to the weight differences of the pre-dried samples further dried at 103 ± 2 °C to a constant weight. The crude protein was determined by the Kjeldahl method on a KjelROC Analyzer, LiquidLine (Furulund, Sweden). A factor of 6.25 was used to recalculate the nitrogen content to the content of nitrogen substances. The crude fiber was determined by gravity on an ANKOM Fiber Analyzer as a non-hydrolyzable sample residue after the hydrolysis in the acid detergent (H_2_SO_4_) and in the alkaline detergent (NaOH) after the subtraction of the ash content in this residue. The crude fat was detected on an extraction device by the Soxhlet method (minimum up to 10 reflux cycles per hour). All fat was extracted from the sample using an extractant (diethyl ether). The ash content was determined by the gravity as the remainder of the sample after the incineration to constant weight at a temperature of 550 ± 20 °C under the prescribed conditions. Acid detergent lignin was determined on the ANKOM device as the mass balance after the cellulose dissolution and other organic substances contained in acid detergent fiber by the action of 72% H_2_SO_4_ [15]. The content determination of nitrogen-free extractives (NFE) was calculated using the following formula: NFE = dry matter (g/kg) − [NS content (g/kg) + fiber content (g/kg) + fat content (g/kg) + ash content (g/kg)].

### 2.4. Determination of Ca and P

Determination of Ca and P was carried out according to Mehlich III. The contents of Ca and P were determined in the samples of feedstuffs and feces extracted by solution prepared as follows: one liter of the solution: 20 g ammonium nitrate, 4 mL ammonium fluoride–EDTA solution (per 500 mL: 69.45 g ammonium fluoride and 36.75 g EDTA, mixed at 50 °C), 11.5 mL nitric acid, and 11.5 mL acetic acid. Samples (10 g) were extracted with 100 mL of the extraction solution under shaking for 10 min. The obtained suspension was filtered through paper filter. The filtrate was analyzed. The spectral determination of phosphorus was carried out on apparatus UNICAM 8625 (Unicam, Cambridge, Great Britain). The content of receivable calcium was determined by using of atomic absorption spectrophotometer PU 9200X (Philips, the Netherlands) in acetylene-air flame with deuterium background correction [16].

### 2.5. Nutrient Digestibility

The nutrient digestibility coefficients were determined by the indicator method using the calculation, as can be seen in the formula below. The indigestible component of the feed dose-lignin, which is the original component of the received feed, was used as an indicator.

Nutrient digestibility coefficient (%) = 100 − [100 × (indicator concentration in feedstuff/indicator concentration in feces) × (nutrient concentration in feces/nutrient concentration in feedstuff)]. 

Digestible energy (DE) was calculated from the values of digestible nutrients according to the equation [17].

DE (MJ) = 0.023 × digestible crude protein (%) + 0.0381 × digestible fat (%) + 0.0172 × digestible fiber (%) + 0.0172 digestible NFE (%).

### 2.6. Analysis of Flavonolignans from Feedstuff and Feces

One gram of homogenized feed or feces was combined with 10 mL of the extraction mixture (H_2_O/ACN/MeOH in the ratio 10:50:40, v/v/v); the sample in the presence of the extraction mixture was vortexed for 1 min, sonicated in an ultrasonic bath for 30 min, and then macerated for 12 h at room temperature in the dark. After 1 min of vortexing, 1 mL of the sample in the extraction mixture was collected and centrifuged for 5 min at 12,000× *g* at room temperature. A total of 200 µL of supernatant was mixed with 400 µL of mobile phase A (MeOH/H_2_O/CH_3_COOH in the ratio 37:63:0.5, v/v/v), vortexed, and centrifuged again. A total of 10 µL of final mixture was injected into an HPLC column. All analyses were carried out in triplicate [18]. The chromatographic system with UV–VIS detection (Knauer, Germany) consisted of solvent delivery pump and chromatographic column RP C18 (Nucleosil C18; 250 mm × 3.0 mm, 5 µm, Phenomenex, USA). Column was termostated at ambient temperature. The UV detection was carried out at 288 nm. Methanol (ACS-grade, Sigma-Aldrich, USA) was used as a mobile phase B. Flow rate of mobile phase was 0.5 mL/min as optimum for analysis. The sample (10 μL) was injected using manual injection. The data obtained were treated by Clarity software (Data Apex, Czech Republic). 

### 2.7. Plasma Profile

Blood samples were taken from the external jugular vein into a heparin plasma preparation tube (Li-Heparin) on days of 20, 41, 62, 83, and 104 regularly for about 2 h after the morning feeding. The blood samples were placed in a centrifuge immediately after the sampling and centrifuged at 3200× *g* at room temperature for 10 min. Non-centrifuged “whole” blood samples were separately pipetted for gamma glutathione peroxidase determination. Blood and plasma samples were stored in tubes (Eppendorf, Germany) and frozen (−20 °C) until the laboratory processing. Total protein, albumin, ALT, AST, ALP, GGT, bilirubin, total cholesterol, HDL, LDL, TAG, NEFA, creatine kinase, creatinine, urea, GSH-Px, TAS, and glucose was determined. Photometric analysis of biochemical blood parameters was carried out by KONELAB T20xt automatic analyzer (Thermo Fisher Scientific, Finland) using commercial kits (Biovendor-Laboratory Medicine, Czech Republic).

### 2.8. Data Evaluation and Statistics

Basic statistical parameters (the mean and standard deviation) were performed and calculated in Microsoft Excel. The obtained data were statistically processed in the program STATISTICA.CZ (12.0) using one-way analysis of variance (ANOVA) and Scheffe’s Test. A value of *p* < 0.05 was considered as statistical significance. Tables and charts were processed in Microsoft Office Excel 2018.

## 3. Results

### 3.1. Nutrient Digestibility

Nutrient digestibility is summarized in Table 4. All horses received the same dry matter intake with respect to age and weight. The season was an autumn/winter. These values correspond to the recommended dry matter intake given by the NRC [14]. Average values of digestible energy have growing tendency (not statistical). The highest values of digestibility of all nutrients occurred at doses of 400 and 700 g milk thistle seed per day. The NFE digestibility ranged from 48.47 ± 1.06% to 59.12 ± 0.89%. Milk thistle seed cakes contain more crude fat, crude protein, and crude fiber, as well as less NFE when compared to barley (Table 2). The mean NFE digestibility was recorded up to 54.91 ± 5.38%. The average value of the crude fat digestibility coefficient reached up to 30.32 ± 14.79%. A statistically significant difference (*p* < 0.05) was found out in crude protein digestibility between pure barley FD (0 g/day) and highest dose of milk thistle seed cakes FD (700 g/day). A statistically significant difference (*p* < 0.05) was found out in crude fiber digestibility between the FD containing up to 200 g of milk thistle seed cakes (the average digestibility of crude fiber was the lowest—11.52 ± 2.21 %) and the FD containing up to 400 g of milk thistle seed cakes (the highest average digestibility of crude fiber was 23.70 ± 0.44%). The amount of Ca and P met the standards [14] in the feed doses for horses. Ca digestibility ranged from 76.34 ± 2.84% (up to 100 g) to 84.89 ± 0.66% (up to 700 g), with a statistically significant difference only between these two groups. Daily FD ranged between 82 and 85.9 g in direct proportion to the dose of milk thistle seed cakes. In contrast, phosphorus digestibility ranged from 6.02 ± 5.49% (up to 100 g) to 37.97 ± 2.56% (up to 400 g), with a statistically significant difference between these two groups (*p* < 0.05). The average digestibility of crude ash ranged from 26.63 ± 4.78% (up to 200 g) to 50.66 ± 4 16% (up to 700 g). A statistically significant difference was found out between these two groups.

### 3.2. Silymarin Digestibility

Silymarin complex digestibility is summarized in Table 5. It is obvious that there was tendency towards growing in all flavonolignans from dose of 100 to 400 g milk thistle seed cakes and stagnation between dose of 400 and 700 g milk thistle seed cakes per day. A statistically significant difference was observed between the minimum dose of milk thistle seed cakes (up to 100 g) in the range of 26.5 ± 21.9% and all other doses regarding the total silymarin digestibility. The digestibility of this dose was approximately about 2–3 times lower compared to all other higher doses. Similar results were found in the individual flavonolignans. The average digestibility of flavonolignans increased with the dose of milk thistle seed cakes. 

### 3.3. Plasma Biochemical Profile

Plasma biochemical profile is summarized in Table 6. Statistically significant differences between mares fed by increasing dose of milk thistle seed cakes were found only for ALT (dose 0 g and 200 g per day) and creatinine (dose of 100 g and 400 g per day).

## 4. Discussion

The observed digestibility of crude fat, NFE, and P corresponded to the results of other authors [19]. Digestibility of a crude protein and Ca was slightly higher compared to the results of Pagan et al. [20]. A slightly lower digestibility was found only in crude fiber [19] compared to our results. The digestibility results of NFE and crude fiber showed similar tendencies, as described by Pagan et al. [19]. This fact suggests that crude fiber in FD reduced the NFE digestibility. Low digestibility was caused perhaps by a higher proportion of crude fiber with a higher proportion of indigestible ADL and lignin that probably negatively affected the digestibility of crude fat and NFE. When comparing the digestibility between 0 and 700 g/day, we found that the digestibilities of crude protein, crude fat, and Ca were significantly higher at 700 g/day milk thistle seed cakes. 

The highest proportion was silybin (up to 61.2%) in the silymarin complex, corresponding to the range of 50–70% [12]. The percentage of individual types of flavonolignans in the silymarin complex contained in the feeding seed cakes is clearly shown in Table 5. Due to the highest proportion of the silymarin complex, the other studies focus most on the bioavailability of silybin. The experimental monitoring of silybin bioavailability was performed in the dependence on daily feed dose, wherein bioavailability grew with daily dose [21]. According to our results (Table 5), digestibility grew with daily dose (up to 400 g). A slight decrease was detected in digestibility with the highest dose (up to 700 g) (without statistical significance) for the main flavonolignans. The most suitable daily dose seemed to be 400 g of milk thistle seed cakes per day according to the digestibility of flavonolignans in the evaluation daily dose of milk thistle seed cakes. To our knowledge, no study has been published on the digestibility of silymarin components by horses. There is, however, a similar study for cows [22], but cows and horses have diametrically opposite gastrointestinal tracts. 

Our results show that the average apparent silybin digestibility increased with a gradual increase of milk thistle seed cakes in the feed dose (statistically significant differences). An interesting finding is that silybin B had average higher digestibility compared to silybin A; however, the proportion of silybin B was detected as being higher in comparison with silybin A in the seed cakes. The determination of the true silybin digestibility is more difficult due to entero-hepatic circulation in which silybin glucuronides are formed in the liver and together with bile are transported to the intestinal tract to be microbially cleaved. Silybin aglycones are released and can be re-absorbed [12]. More than 20–50% of silymarin is absorbed by oral feeding, as has been reported. About 80% is excreted in the bile and about 10% enters the entero-hepatic circulation [21]. Silybin occurs in the form of glucuronide and sulphate conjugates in bile (assuming the silybin use is in the range of 20–40% in bile and the remainder is excreted in the feces), as reported in the description of silybin metabolism [12].

The trend of growing silymarin digestibility with a higher dose of milk thistle seed cakes is very interesting. This phenomenon could be caused by growing content of crude fat in dry matter (Table 3), which could influence solubility of silymarin ([12,23]) complex due to relative high fat digestibility by horses with growing content of crude fat in feed dose (a similar trend is shown in Table 4). From this point of view, the crude fat content in dry matter in the monitored feed doses can be considered as feed doses with lower content of crude fat (range from 1.97% to 2.39% crude fat in dry matter—see Table 3) because sport horses can be fed doses with 16% (160 g/kg dry matter) crude fat [24]. On the other hand, this could mean that the microbiome fully adapted to the digestion of silymarin until after 9 weeks (possible reason for stagnation in digestibility values between 400 and 700 g; Table 5). Further research is needed. According to recent findings, silymarin has been shown to modulate the gut microbiota [25]. However, Valentova et al. [26] state that the flavonolignans of humans, especially at higher (pharmacological) doses, are relatively resistant to biotransformation by gut microbiota (also a possible factor of stagnation in digestibility values between 400 and 700 g; Table 5). The question still remains about the mechanism of silymarin complex digestibility in horses. 

All monitored biochemical parameters of plasma were in the reference range of values. ALP values of all horses were outside the range given by Doubek [27], which is without diagnostic significance. The exceptions were reduced cholesterol levels at the doses of 0 and 200 g and reduced total protein level at the 200 g dose with physiological albumin level. Horse nutrition can have an impact on the health and lifespan of horses, but this fact requires further research. Nutritional intervention proves the potential of silymarin complex to prevent the development of inflammatory processes [28]. The feeding of milk thistle seed cakes offers a significant preventive application with its antioxidant and hepatoprotective effects in horse nutrition not only for sports purposes. The effects of silymarin against the development of laminitis or metabolic syndrome are also known in horses [10]. In addition, a negative correlation has been found between PUFA and pro-inflammatory cytokines [29]. The horses received a sufficient amount of protein assessed by determining the plasma concentration of urea, albumin, and the total protein according to the reference range of values (urea, total protein, and albumin) [30]. Urea and albumin values were detected as normal in sample 3 (up to 200 g), with lower concentration of the total protein being found. However, horses received a high ratio of calcium and phosphorus that was not entirely ideal, and plasma values were detected physiologically ([27,30]).

Statistically significant differences between groups (different doses of milk thistle seed cakes) were found only for creatinine levels, which was directly related to physical exercise. The values were physiological, and thus creatinine values did not indicate a “better” or “worse” condition of the kidneys. Decreased creatinine and urea were observed after the silymarin feeding in induced renal impairment (higher urea, creatinine) in other studies [31].

No significant difference was found between the doses of milk thistle seed cakes in lipid intake parameters. On average, the highest (insignificant) values of the total cholesterol were detected in HDL and LDL (but not TAG) in the fourth sample at a daily dose of 400 g, with the highest observed silymarin digestibility. The feeding of milk thistle seed cakes, containing the rest of the oil, did not increase the TAG level in the blood of horses, which is a desirable phenomenon because higher TAG levels are characteristic for horses at risk of metabolic syndrome [32].

Regarding the indicators of oxidative stress such as TAS and GSH-Px, no significant difference was observed. The highest digestibility of the silymarin complex was found at a dose of 400 g/day, and also the average highest values of antioxidants were determined.

In the plasma of horses, the level of flavonolignans was determined below the detection limit for all doses of milk thistle seed cakes (data not shown). The silybin bioavailability, based on the results of Hackett et al. [33], should reach <1% in horses, and no accumulation occurred with twice-daily dosing for 7 days (silybin did not show toxic effects, even at high doses).

## 5. Conclusions

Milk thistle seed cakes are an advantageous source of energy and well-digestible nutrients (higher digestibility of protein, fat, Ca, P) in horse nutrition. An increase in protein digestibility (the highest protein digestibility in the range of 81.27 ± 1.00% at the highest dose of 700 g/day), fat (in the range of 46.34 ± 3.19%), calcium (in the range of 84.89 ± 0.66%), and phosphorus (in the range of 32.55 ± 6.32%) was detected in the average values of DE (seed cakes contained more fat compared to barley), together with an increasing doses of milk thistle seed cakes. An interesting result was found in the determination of the silymarin complex digestibility and its individual components. The silymarin digestibility was significantly increased with doses up to 400 g/day, which means that the digestibility reached the highest level (in the range of 77.4 ± 4.6%). The digestibility stagnated (in the range of 70.6 ± 3.2%) at the dose of 700 g/day. The highest average digestibility was processed at dose of 400 g/day, which could therefore be the most suitable dose from this point of view.

A statistically significant difference between plasma biochemical values was detected only for creatinine values. All horses were clinically healthy and showed no pathological findings, and thus the positive effects of silymarin could not be seen. In the plasma of horses, the level of flavonolignans was determined below the detection.

This phenomenon of silymarin digestibility could be caused by growing content of crude fat in dry matter that could influence solubility of silymarin complex due to relative high fat digestibility by horses with growing content of crude fat in feed dose; the next factor could be that silymarin administration tended to change the microbiota diversity, and the third theory is that silymarin could be relatively resistant to biotransformation by gut microbiota at higher doses. A combination of these factors is possible as well. Further research for clarifying this phenomenon is necessary. 

## Figures and Tables

**Table 1 animals-11-01687-t001:** The scheme of daily feed doses and intake during the experiment.

The Day of the Exp.	Barley Granules with a Proportion of Milk Thistle Seed Cakes *					Alfalfa Hay (kg/day)	Rye Silage (kg/day)	* Dose of Ext. Barley (kg/day)	* Dose of Milk Thistle Seed Cakes (g/day)
	0%	5%	10%	20%	35%				
0–20	2 kg	-	-	-	-	5 kg	5 kg	2 kg	-
21–41	-	2 kg	-	-	-	5 kg	5 kg	1.9 kg	100 g
42–62	-	-	2 kg	-	-	5 kg	5 kg	1.8 kg	200 g
63–83	-	-	-	2 kg	-	5 kg	5 kg	1.6 kg	400 g
84–104	-	-	-	-	2 kg	5 kg	5 kg	1.3 kg	700 g

**Table 2 animals-11-01687-t002:** The nutrient composition of feed forming the basic feed dose of horses.

	Alfalfa Hay	Barley Granules with a Proportion of Milk Thistle Seed Cakes					Rye Silage
		0%	5%	10%	20%	35%	
Dry matter (g)	933.0	888.0	884.0	879.0	870.0	894.0	330.0
Crude fiber (g)	251.8	34.4	44.9	55.5	76.5	99.5	117.0
Crude fat (g)	15.6	20.1	23.0	25.9	31.7	37.1	8.2
Crude protein (g)	155.2	114.4	118.8	123.2	131.9	143.0	32.8
Crude ash (g)	103.1	23.4	25.2	27.0	30.6	34.6	41.5
Adl-lignin (g)	61.9	7.7	12.1	16.5	25.3	36.5	18.8
Ca (g)	13.9	0.7	1.0	1.35	2.0	2.6	2.1
P (g)	1.7	3.1	3.3	3.4	3.8	4.4	0.7
Silymarin (g)	-	-	1.7	3.4	6.7	9.7	-

**Table 3 animals-11-01687-t003:** The nutrient composition of feed with different proportions of milk thistle seed cakes.

	Dry Matter (kg)	Crude Protein (g/kg)	Crude Fat (g/kg)	Crude Fiber (g/kg)	NFE (g/kg)	Crude Ash (g/kg)	Ca (g/day)	P (g/day)	Silymarin (g/day)
0 g/day	8.16	144.0	19.7	237.3	503.5	95.0	82.0	18.5	0
100 g/day	8.16	145.0	20.5	240.1	498.2	96.0	82.9	18.8	3.4
200 g/day	8.14	146.0	21.2	243.0	492.3	96.4	83.5	19.1	6.8
400 g/day	8.12	149.0	22.7	248.7	482.0	97.5	84.8	19.7	13.4
700 g/day	8.17	150.9	23.9	252.9	474.4	97.9	85.9	21.1	19.4

**Table 4 animals-11-01687-t004:** Nutrient digestibility (%) depending on daily doses of milk thistle seed cakes.

Group	DE (MJ)	Crude Protein (%)	Crude Fat (%)	Crude Fiber (%)	NFE (%)	Crude Ash (%)	Ca (%)	P (%)
	Average ± SE	Average ± SE	Average ± SE	Average ± SE	Average ± SE	Average ± SE	Average ± SE	Average ± SE
**0 g/day**	4.00 ± 0.25	75.54 ± 0.67 ^bc^	18.86 ± 5.78 ^b^	21.85 ± 1.67 ^ab^	58.25 ± 1.09 ^abd^	45.55 ± 3.88 ^ab^	76.34 ± 2.84 ^b^	10.87 ± 6.02 ^bd^
**100 g/day**	4.13 ± 0.30	75.78 ± 0.86 ^c^	25.98 ± 11.80 ^ab^	23.71 ± 2.05 ^b^	59.12 ± 0.89 ^a^	26.63 ± 4.78 ^c^	78.39 ± 0.88 ^ab^	6.02 ± 5.49 ^bc^
**200 g/day**	4.14 ± 0.11	74.50 ± 0.72 ^c^	36.52 ± 2.83 ^ab^	11.52 ± 2.21 ^c^	48.47 ± 1.06 ^c^	29.00 ± 2.44 ^bc^	81.19 ± 0.41 ^ab^	17.73 ± 1.92 ^abc^
**400 g/day**	4.13 ± 0.26	79.26 ± 1.28 ^ab^	23.90 ± 5.35 ^ab^	23.70 ± 0.44 ^ab^	57.66 ± 1.39 ^ab^	41.92 ± 2.11 ^abc^	82.00 ± 1.32 ^ab^	37.97 ± 2.56 ^a^
**700 g/day**	4.76 ± 0.17	81.27 ± 1.00 ^a^	46.34 ± 3.19 ^a^	14.51 ± 2.58 ^abc^	51.06 ± 2.56 ^bc^	50.66 ± 4.16 ^a^	84.89 ± 0.66 ^a^	32.55 ± 6.32 ^ad^

SE: standard error; ^a:b:c:d^ the averages of the same order marked by different letters are significantly different from each other (*p* < 0.05).

**Table 5 animals-11-01687-t005:** Digestibility of silymarin and the individual types of flavonolignans (%).

Group	Content in %	100 g/day	200 g/day	400 g/day	700 g/day
	Average ± SE	Average ± SE	Average ± SE	Average ± SE	Average ± SE
**Silychristine**	25.5	35.7 ± 8.9 ^c^	59.9 ± 3.3 ^b^	83.6 ± 1.9 ^a^	78.3 ± 1.2 ^ab^
**Silydianin**	0.7	47.5 ± 9.4 ^b^	62.1 ± 2.8 ^ab^	75.1 ± 2.9 ^ab^	85.8 ± 1.3 ^a^
**Silybin A**	24.4	30.5 ± 11.4 ^b^	49.1 ± 3.4 ^ab^	76.9 ± 2.2 ^a^	67.9 ± 1.6 ^a^
**Silybin B**	36.8	50.9 ± 8.5 ^b^	70.2 ± 4.3 ^ab^	80.1 ± 2.7 ^a^	69.3 ± 1.3 ^ab^
**Isosilybin A**	9.7	10.4 ± 10.4 ^b^	35.2 ± 3.7 ^b^	71.1 ± 2.7 ^a^	68.4 ± 2.4 ^a^
**Isosilybin B**	2.9	27.4 ± 7.8 ^b^	37.2 ± 8.7 ^ab^	70.9 ± 5.6 ^a^	65.9 ± 6.6 ^a^
**Silymarin complex**	100	26.5 ± 21.9 ^b^	56.9 ± 7.3 ^a^	77.4 ± 4.6 ^a^	70.6 ± 3.2 ^a^

SE: standard error; ^a:b^ the averages of the same order marked by different letters are significantly different from each other (*p* < 0.05).

**Table 6 animals-11-01687-t006:** Plasma profile.

Group	0 g/Day	100 g/Day	200 g/Day	400 g/Day	700 g/Day
	Average ± SE	Average ± SE	Average ± SE	Average ± SE	Average ± SE
**Total protein** (g/L)	53.89 ± 3.42	57.18 ± 3.88	49.52 ± 2.35	60.90 ± 2.87	54.18 ± 1.75
**Albumin** (g/L)	31.90 ± 2.40	32.66 ± 2.12	28.62 ± 1.87	36.89 ± 0.75	31.83 ± 1.06
**ALT** (µkat/L)	0.07 ± 0.01 ^b^	0.08 ± 0.01 ^ab^	0.11 ± 0.02 ^a^	0.08 ± 0.01 ^ab^	0.08 ± 0.01 ^ab^
**AST** (µkat/L)	3.23 ± 0.18	3.57 ± 0.18	3.97 ± 0.36	3.98 ± 0.22	3.58 ± 0.22
**ALP** (µkat/L)	2.22 ± 0.18	2.01 ± 0.26	2.23 ± 0.26	2.08 ± 0.30	1.74 ± 0.27
**GGT** (µkat/L)	0.24 ± 0.06	0.28 ± 0.06	0.25 ± 0.04	0.26 ± 0.03	0.22 ± 0.02
**Bilirubin** (µmol/L)	11.72 ± 1.33	12.13 ± 3.04	10.68 ± 0.46	12.77 ± 0.66	11.53 ± 1.85
**Cholesterol** (mmol/L)	1.58 ± 0.16	1.86 ± 0.23	1.55 ± 0.11	2.07 ± 0.10	1.88 ± 0.08
**HDL-chol.** (mmol/L)	0.95 ± 0.11	1.03 ± 0.08	0.89 ± 0.06	1.17 ± 0.06	1.10 ± 0.03
**LDL-chol.** (mmol/L)	0.64 ± 0.06	0.67 ± 0.05	0.65 ± 0.07	0.88 ± 0.06	0.78 ± 0.06
**TAG** (mmol/L)	0.21 ± 0.03	0.19 ± 0.04	0.28 ± 0.06	0.25 ± 0.03	0.29 ± 0.03
**NEFA** (mmol/L)	0.12 ± 0.01	0.11 ± 0.02	0.13 ± 0.02	0.13 ± 0.00	0.15 ± 0.01
**CK** (µkat/L)	3.17 ± 0.15	2.53 ± 0.35	3.56 ± 0.37	2.92 ± 0.22	2.58 ± 0.19
**Creatinine** (µmol/L)	118.93 ± 3.41 ^ab^	106.25 ± 7.03 ^b^	113.95 ± 7.16 ^ab^	127.04 ± 3.26 ^a^	135.55 ± 4.18 ^ab^
**Urea** (mmol/L)	6.51 ± 0.29	6.74 ± 0.70	6.16 ± 0.22	6.27 ± 0.36	6.39 ± 0.51
**GSH-Px** (µkat/L)	933.86 ± 100.60	860.50 ± 69.34	927.29 ± 56.80	1085.99 ± 95.52	1081.45 ± 75.60
**TAS** (mmol/L)	0.92 ± 0.04	0.92 ± 0.06	0.93 ± 0.04	1.05 ± 0.06	0.92 ± 0.04
**Glucose** (mmol/L)	3.78 ± 0.12	4.80 ± 0.71	3.90 ± 0.39	4.05 ± 0.16	4.04 ± 0.11
**Ca** (mmol/L)	2.13 ± 0.09	2.44 ± 0.24	2.24 ± 0.11	2.72 ± 0.08	2.59 ± 0.11
**P_i_** (µkat/L)	0.87 ± 0.14	0.82 ± 0.14	0.84 ± 0.08	0.81 ± 0.10	0.80 ± 0.12

SE: standard error; ^a:b^ the averages of the same order marked by different letters are significantly different from each other (*p* < 0.05).

## Data Availability

Data is contained within the article.

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
