# Peer review of "Dose Effect of Milk Thistle (Silybum marianum) Seed Cakes on the Digestibility of Nutrients, Flavonolignans and the Individual Components of the Silymarin Complex in Horses"

_animals, 2021, doi:10.3390/ani11061687_

Round 1
Reviewer 1 Report
The Authors evaluate digestibility of silymarin complex and the effect of nutrient digestibility in horses.
The investigated topic is very interesting.
The manuscript is well written, presented and discussed.
In general, the structure of the article is satisfactory and in agreement with the journal instructions for authors.
The objectives of the paper are of interest and fit well within the scope of the journal.
In my opinion, the manuscript could be accepted for publication in Animals.
Author Response
Dear reviewer, thank you for your expert comments. Sincerely, Hana Dočkalová
Reviewer 2 Report
The topic of this manuscript fits the scope of the journal, however the experimetnal design is not adequate for the research questions. Authors intended to assess digestibility of milk thistle in horses, but the treatments are not independent of each other, as the same animal received all the doses following each other, without washout periods. The microbiome is never the same when an animla changes from one treatment to the other, impacting digestive physiology and efficacy. In this regard, the statistical model used is also not appropriate to the experimental design (no independent treatments, required for ANOVA, and measurements done on the same animals = repeated measures).
Feed intake is not shown and this can also have an impact on digestibility of feed ingredients.
Introduction is too extensive and only at the end it is clear why milk thistle is given to horses and under which circumstances. Additionally to this comment, the physiological status of the mares is also not mentioned. were they pregnant?
The experimental design is also not correct to test the possible role of the feed additive as an antioxidant. there was no challenge to trigger an oxidative reaction in the mares, so as to assess the protective role of milk thistle.
The discussion section is also weak and not to the point.
Author Response
Dear reviewer, I have taken your comments into account and edited the article. Explanations of comments and remarks are written in red in the annex to the answer. Thank you for your expert comments. Sincerely, Hana Dočkalová

Reviewer 3 Report
Please correct the marked mistake!

Author Response

(The authors gave the same response as above.)

Round 2
Reviewer 2 Report
The authors still did not clarify what the research question of this study was. At the end of the introduction authors talk about digestibility of the seed cakes assessment, but the majority of the results and discussion focus on the blood profile of inflammation and oxidative stress. Authors need to revise the research question and write it clearly at the end of the introduction.
Another point is that authors use the words 'digestibility' and 'bioavailability' through each other, when these words ahev very specific and different meanings. This needs to be clarified and not be mixed up! and make it clear if the digestibility affects availability or not of the metabolites that then have an anti-inflammatory or anti-oxidative role.
the length of the wash-out period is too small to allow a full recovery of the microbiome. authors shoudl test for carry-over effects, by adding period as a factor to their statistical model and look for the interactions with factor period! or use repeated measures as it seems that it was a kind of Latin square design that was used. if this is the case, then it should be described as such and evaluateed as such!
if mares were not exposed to physical activity, how is it possible to make a good conclusion regarding the potential of the seed cakes to make the animal more robust to diseases/inflammation? the animals were not challenged. they were at 'neutral/resting' status. this is not enough to conclude on the potential anti-inflammatory or anti-oxidative role of the seed cakes! further, the fact that there were no physical activity or stress, then what is the meaning of an improved digestibility of the seed cakes/feed to the animals. it will not bring any added value to the animal, if the animal does not need the extra nutrients. these will just be excreted in the feces, into the environment!
How do authors explain the increased 'digestibility' of the seed cakes with increasing doses of the seed cake? would not be expected to be the same digestibility of the seed cake? and from where do the extra ADL and crude fibre come from in the higher dose treatments? authros were replacing only extruded barly by the seed cake. does it mean that your seed has so much lignin? but then you would expect lignin and crude fibre to have a negative impact on overal digestibility of the feed and of the different nutrients! what is even worse is that lignin was used as indicator to assess digestibility, but if this is not constant throughout the different treatments, then digestibility is not assessed correctly! authors shoudl look for another indicator to assess digestibility of the feed and of the seed cake!
English language is also an issue as the sentences, particularly ion the discussion are hard to read and follow what the authors are trying to say.
Author Response
Thank you for pointing out possible ambiguity when using the terms bioavailability and digestibility, we have modified the article for better comprehensibility. Biochemical blood parameters for health assessment and are an integral part of the study. There seem to be a lot of results in biochemistry because we wanted to have a comprehensive picture. The health status of the liver is expressed mainly by the values of enzymes in the cell membrane of hapatocytes and bilirubin, the health status of the kidneys creatinine and urea, the metabolism of lipids TAG, NEFA, cholesterol, the metabolism of total protein proteins, albumin and urea. Blood counts were an important objective indicator of health. If mares were exposed to stress, the antioxidant / anti-inflammatory effects of silymarin could probably be better manifested. In the organism itself, there are constant oxidation-reduction processes. In further research, we want to focus on the physical load of horses and the effect of silymarin. Thank you for your factual suggestion for the methodology, when recent results of other authors suggest the effect of silymarin on the intestinal microbiota. We statistically processed the results using the SNEDECOR and COCHRAN methodology, using the Scheffeé tests test. In further research, we will use the methodology of Latin square design and wash-out periods to better clarify our results.